# Extreme neonatal hyperbilirubinaemia in refugee and migrant populations: retrospective cohort

Eva Maria Nadine Wouda  ,[1,2] Laurence Thielemans,[1,3] Mue Chae Darakamon,[1] Aye Aye Nge,[1] Wah Say,[1] Sanda Khing,[1] Borimas Hanboonkunupakarn,[4] Thatsanun Ngerseng,[4] Jordi Landier,[1,5] Patrick Ferry van Rheenen,[6] Claudia Turner,[4,7,8] Francois Nosten,[1,8] Rose McGready,[1,8] Verena Ilona Carrara [1,8,9]

## ABSTRACT

**Objective** To describe neonatal survival and long-term neurological outcome in neonatal hyperbilirubinaemia (NH) with extreme serum bilirubin (SBR) values.

**Design** Retrospective chart review, a one-off neurodevelopmental evaluation.

**Setting** Special care baby unit in a refugee camp and clinics for migrant populations at the Thailand–Myanmar border with phototherapy facilities but limited access to exchange transfusion (ET).

**Patients** Neonates ≥28 weeks of gestational age with extreme SBR values and/or acute neurological symptoms, neurodevelopment evaluation conducted at 23–97 months of age.

**Main outcome measures** Neonatal mortality rate, prevalence of acute bilirubin encephalopathy (ABE) signs, prevalence of delayed development scores based on the Griffiths Mental Development Scale (GMDS).

**Results** From 2009 to 2014, 1946 neonates were diagnosed with jaundice; 129 (6.6%) had extreme SBR values during NH (extreme NH). In this group, the median peak SBR was 430 (IQR 371–487) µmol/L and the prevalence of ABE was 28.2%. Extreme NH-related mortality was 10.9% (14/129). Median percentile GMDS general score of 37 survivors of extreme NH was poor: 11 (2–42). 'Performance', 'practical reasoning' and 'hearing and language' domains were most affected. Four (10.8%) extreme NH survivors had normal development scores (≥50th centile). Two (5.4%) developed the most severe form of kernicterus spectrum disorders.

**Conclusion** In this limited-resource setting, poor neonatal survival and neurodevelopmental outcomes, after extreme NH, were high. Early identification and adequate treatment of NH where ET is not readily available are key to minimising the risk of extreme SBR values or neurological symptoms.

## What is known about the subject?

► Extreme values of serum bilirubin (SBR) in the neonatal period can lead to death and long-term disability.
► Subtle disabilities and neurodevelopment delay might occur after recovery of neonatal hyperbilirubinaemia.
► Low-income and middle-income countries bear the highest burden of disease but have limited access to diagnosis, treatment and supportive care for disability.

## What this study adds?

► A large majority of neonates with extreme SBR levels survive after on-site phototherapy but limited access to exchange transfusion.
► Long-term neurodevelopmental impairments, more pronounced in 'performance', 'reasoning skills' and 'language', affect survivors already bearing a heavy burden of chronic malnutrition and infections.
► Early recognition and adequate treatment are key to minimising the risk of extreme SBR values or neurological symptoms.

For numbered affiliations see end of article.

**Correspondence to**
Dr Verena Ilona Carrara; verena@shoklo-unit.com

## BACKGROUND

In the most severe form of neonatal hyperbilirubinaemia (NH), serum bilirubin (SBR) levels rise to extreme values, causing a life-threatening condition that can lead to lifelong disability.[1] Acute bilirubin encephalopathy (ABE) happens during the early phase of the disease, producing a vast array of neurological symptoms from mild and reversible to severe and irreversible. Its physiopathology is complex; however, prematurity, sepsis, glucose-6-phosphate dehydrogenase (G6PD) deficiency and ABO/rhesus incompatibility contribute to its development.[2–5] Neurological sequelae might progress to kernicterus spectrum disorders (KSD) ranging from subtle to severe motor and/or auditory dysfunction,[6] as well as language delay, attention disorders or lower executive function capacities.[7]

A review of the worldwide burden of severe NH published in 2017 reported an incidence of 251.3 per 10 000 live births in Southeast Asia,

the second highest after Africa.[8] NH-related mortality is high: 1309 deaths per 100 000 live births in 2016.[9] Low-income and middle-income countries bear the heaviest burden; two-thirds of deaths and severe sequelae occur in sub-Saharan Africa and South Asia; 3%–20% of survivors have normal long-term development.[10–13]

Most NH-related research follows the American Academy of Paediatrics guideline for neonates with an estimated gestational age (EGA) of ≥35 weeks, in which SBR levels of >25 mg/dL (428 µmol/L) at age >72 hours are considered extreme and require exchange transfusion (ET).[14] The British National Institute for Health and Clinical Excellence (NICE) guideline proposes dynamic treatment thresholds based on postnatal age and varying with gestational ages from 23 to ≥38 weeks.[15] SBR values considered extreme therefore start as low as 13 mg/dL (230 µmol/L) for the youngest EGA. However, publications reporting outcomes of neonates treated following these dynamic treatment thresholds are still rare.[16]

The Shoklo Malaria Research Unit (SMRU) set up a special care baby unit (SCBU) in 2008 in Mae La, the most populous refugee camp along the Thailand–Myanmar border, to provide basic care for neonates including oxygen, intravenous antibiotics, nasogastric feeding, phototherapy and basic laboratory tests.[17] Starting 2011, another two SCBUs were set up in clinics for migrant populations. Care for neonates with NH improved over time following the implementation of standardised guidelines based on the NICE,[15] systematic heel prick SBR measurements with onsite bilirubinometer (Pfaff Medical Bilimeters 2 and 3) and irradiance level optimisation with the introduction of light emitting diode (LED) phototherapy units.[18] In live births from 2009 to 2014, NH prevalence was 18%; 9% of these neonates had at least one SBR value exceeding the NICE threshold at which ET should be considered.[18] ET is an intervention with potentially serious complications performed in an intensive care unit rather than in a primary health facility.[19 20] Due to the inherent difficulties of sending a refugee or migrant person to a specialised centre, only some neonates could be referred.

The objective of this review was to evaluate the immediate and long-term clinical and neurodevelopmental outcomes of children from a low-resource setting with SBR values rising to levels considered extreme.

## METHODS
Three SMRU SCBU facilities contributed to the data in refugees (1 January 2009–31 December 2014[17]) and in migrants (1 January 2011–31 December 2014) (figure 1).

### Inclusion criteria
Clinical charts of all liveborn singletons of ≥28 weeks EGA without major congenital abnormalities born to mothers attending SMRU antenatal care admitted for a clinical diagnosis of jaundice were reviewed.

### Definitions
For this analysis, NH with SBR values rising above levels that could justify ET, reported in the text as 'extreme NH', was defined as either (1) two consecutive SBR measurements above the ET threshold of the NICE guideline, (2) SBR levels rising faster than 8.5 µmol/L/hour in combination with one SBR measurement above the ET threshold or with clinical symptoms of ABE, or (3) a clinical diagnosis of ABE.[15]

In the absence of a scoring system for neurological symptoms, a simplified ABE definition was used and consisted of two or more symptoms mentioned in the chart ('quiet', 'sleepy', 'less sucking', 'irritable', 'lethargic', 'floppy', 'hypotonic', 'rigid', 'apnoea', 'seizure' and 'kernicterus') occurring within 2 days of the SBR peak.[21]

A computerised formula based on the NICE threshold graphs was used to select charts with SBR measurements fitting inclusion criteria. Clinical charts and originally drawn SBR trajectories were cross-checked to confirm eligibility. The first author and a paediatrician (LT) reviewed clinical charts for classifying available neurological symptoms, collecting ET referral and possible cause of death information.

Three available factors aggravating the risk of ABE other than prematurity (blood group ABO incompatibility, G6PD deficiency measured by fluorescent spot test[22] and neonatal sepsis) were extracted from computerised databases. Clinical diagnoses followed the local consensus-based SMRU neonatal guidelines. EGA of 35 weeks defined prematurity in this cohort as it is commonly used to assess the risk of extreme NH in the literature.[3 12 23 24] In this setting, most pregnancies have EGA calculated by ultrasound[25] or by the best estimate of gestation available if there is late presentation to antenatal care.[26] SBR and G6PD results were provided in near real time by locally trained, routinely quality controlled laboratory technicians.

### Neurodevelopmental assessment
Children with extreme NH discharged alive from SCBU were traced using their parents address. For those consenting to participate, a one-off appointment was organised at the nearest SMRU clinic between November 2016 and March 2017. Medical history extracted from the child's medical booklet and clinical and neurological examination findings were reported on a standardised form. Maternal concern about the child's wellbeing was evaluated by asking whether the child had difficulty seeing, hearing, speaking, bathing, dressing or in terms of mobility, and reported as yes/no. The Griffiths Mental Development Scales—Extended Revised (GMDS-ER[27]) was used to assess neurocognitive development: it spanned the possible age distribution of this cohort and was familiar to the staff who could perform it in their own language. Test items were scored as 'pass' or 'fail' on six individual subscales, and percentile scores were derived from raw scores. A general neurodevelopment score EGA-adjusted was calculated per GMDS-ER

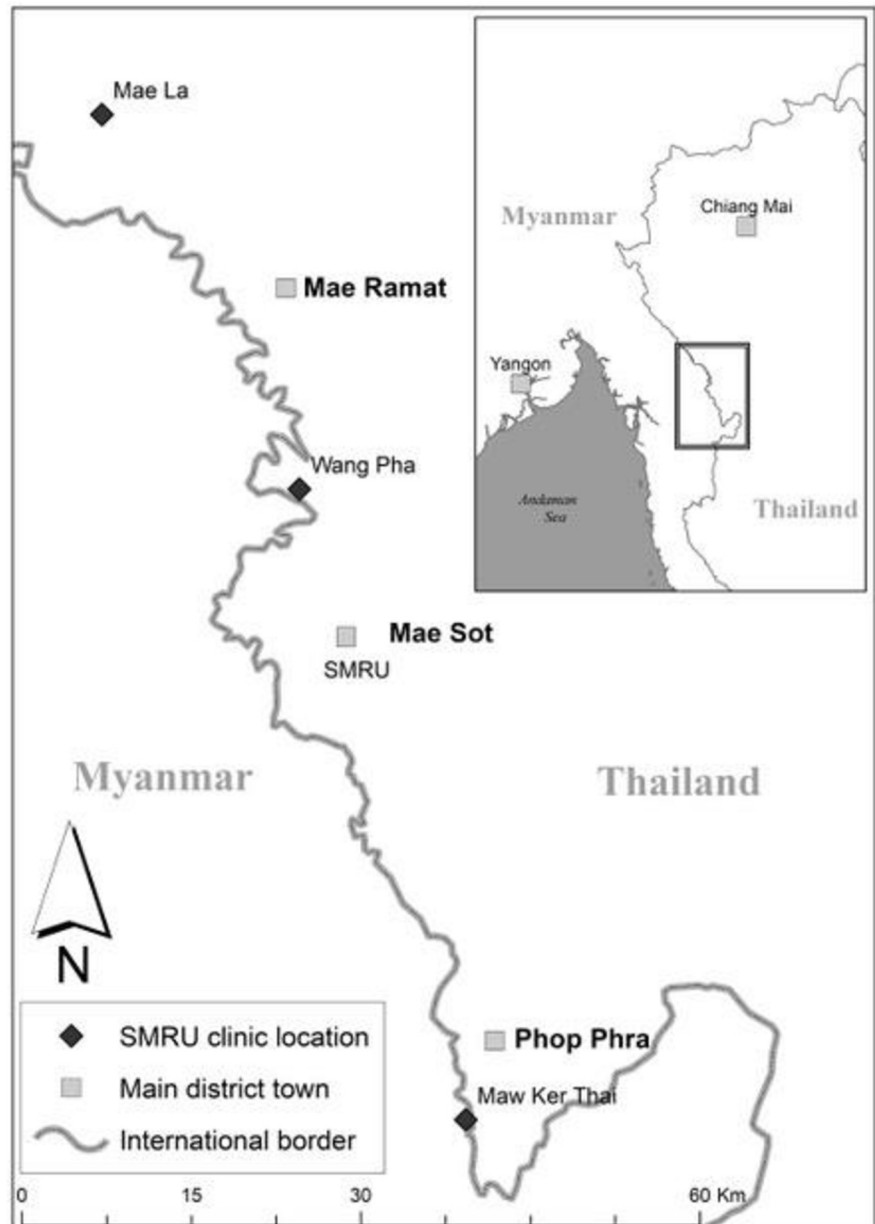

**Figure 1** Location of the three SMRU clinics. The three clinics are Mae La clinic for refugees, Wang Pha clinic and Maw Ker Thai clinic for migrant population. The SMRU main office is located in Mae Sot town. Photo credit: Daniel Parker. SMRU, Shoklo Malaria Research Unit.

administration and analysis manuals and categorised as poor (<10th percentile), delayed (10–49th percentiles) and normal (50th percentile or above). Visual function was assessed with the Cardiff cards for visual acuity and visual contrast.[28] All tests were performed by local staff trained and regularly quality-controlled in clinical neurological examination and neurodevelopmental testing.[29 30]

### Statistics
Analysis was performed using IBM SPSS Statistics for Macintosh V.24.0.0.1. Continuous data were described by their median and IQR and compared using the Mann-Whitney test. Categorical data were described by their proportion and compared with $\chi^2$ test, Fisher's exact test or $\chi^2$ test for trend as appropriate. Factors associated with

ABE were evaluated in univariate analysis and reported as OR with 95% CI.

### Patient and public involvement
The Tak Province Border Community Ethics Advisory Board members representing the local community where the study took place reviewed the protocol and their input into the design of the project was taken into account.

### RESULTS
Between 2009 and 2014, 13538 SMRU antenatal attendees gave birth at ≥28 weeks' EGA to 12963 congenitally normal singletons; 2980 (23.0%) neonates were hospitalised in the SCBU; 1946 were diagnosed with jaundice;

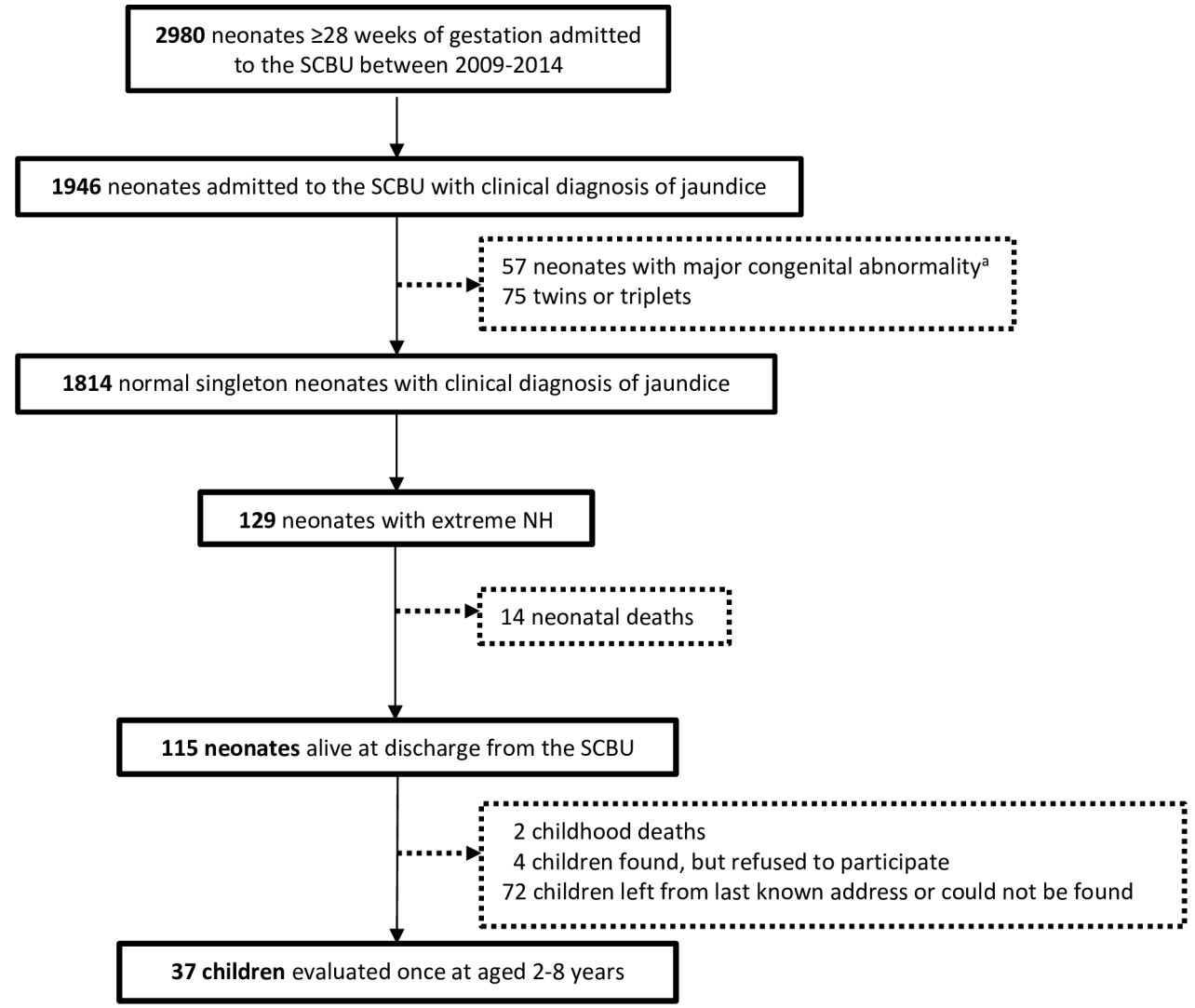

**Figure 2** Flowchart representing the selection of neonates with extreme hyperbilirubinaemia (NH). ᵃMajor congenital abnormality was defined as an abnormal finding on surface examination and/or auscultation of the heart and lungs reported in the medical chart and confirmed by a medical doctor. NH, neonatal hyperbilirubinaemia; SCBU, special care baby unit.

and 129 fulfilled the criteria of extreme NH (figure 2). Extreme NH incidence among neonates delivered in SMRU birthing units (110/10398) was estimated at 105.8 per 10000 live births (95% CI 87.0 to 127.4).

The general characteristics of the 129 neonates are summarised in table 1. The majority (70.5%) were from the refugee camp as expected given the longer data period; almost half (46.5%) were born from primigravid mothers and 37 (28.7%) were <35 weeks' EGA. Sepsis was clinically diagnosed prior to onset of NH in 26.4% of hospitalisations.

The median (IQR) value of the SBR peak was 430 (371–487) µmol/L and occurred at a median of 97 (68–136) hours of life. Five neonates were referred; four successfully underwent ET; one died of sepsis without confirmation of ET being attempted.

### Mortality and ABE
Extreme NH case fatality rate was 10.9%, giving a mortality rate of 108.1 per 100000 live births (95% CI

59.1 to 181.4); neurological symptoms compatible with ABE were reported in 28.2% of the clinical charts of neonates not transferred for ET (35/124). All deaths occurred among neonates with ABE.

The odds of developing ABE was higher among neonates born <35 weeks' EGA (41.6%, 15/36, vs 22.7%, 20/88, OR 2.4 (95% CI 1.1 to 5.7)) or treated for clinical sepsis prior to onset of NH (51.6%, 16/31, vs 20.4%, 19/93, OR 4.2 (95% CI 1.8 to 9.9)) but not for those with blood group ABO incompatibility or G6PD deficiency.

### Long-term outcomes
Thirty-seven children survivors of extreme NH were present for neurodevelopmental assessment, including three who received ET. Their neonatal and maternal characteristics were similar to those of the 78 unavailable or untraceable children (table 2).

Clinical, visual, neurological and neurodevelopmental outcomes are presented in table 3. While none of the children seen were acutely ill, 30.6% reported more than

**Table 1** Characteristics of neonatal period of 129 neonates with extreme NH

| Characteristics | | Extreme NH (N=129) |
|---|---|---|
| Year of admission | 2009 | 22 (17.0) |
| | 2010 | 26 (20.2) |
| | 2011 | 12 (9.3) |
| | 2012 | 24 (18.6) |
| | 2013 | 27 (20.9) |
| | 2014 | 18 (14.0) |
| Status | Refugee | 91 (70.5) |
| | Migrant | 38 (29.5) |
| Gender (male) | | 75 (58.1) |
| Gestational age ≥35 weeks* | | 92 (71.3) |
| Place of delivery | SMRU clinic | 110 (85.3) |
| | Tertiary hospital | 4 (3.1) |
| | Home | 15 (11.6) |
| Birth weight (kg) (n=125), median (IQR)† | | 2.53 (1.96–2.90) |
| Small for gestational age (n=125)‡ | | 23 (18.4) |
| Poor start to life§ | | 15 (11.6) |
| Mother primigravid | | 60 (46.5) |
| Normal vaginal delivery | | 114 (88.4) |
| Age at admission (days), median (IQR) | | 1 (0–4) |
| Duration of admission (days), median (IQR) | | 6 (4–11) |
| Extreme NH category¶ | Two SBRs above ET | 101 (78.3) |
| | Rapid SBR rise+ABE signs | 27 (20.9) |
| | Clinical diagnosis of ABE | 1 (0.8) |
| Duration of phototherapy (hours), median (IQR) | | 76 (48–124) |
| Peak SBR (µmol/L), median (IQR) | | 430 (371–487) |
| Age at peak SBR (hours), median (IQR) | | 97 (68–136) |
| Peak SBR at <72 hours of life | | 41 (31.8) |
| Underwent ET | | 4 (3.1) |
| Potential blood group ABO incompatibility** (n=126) | | 24 (19.0) |
| G6PD-deficient male†† (n=75) | | 33 (44.0) |
| Polycythaemia‡‡ | | 10 (7.8) |
| Birth trauma (visible bruising after birth recorded in the chart) | | 6 (4.7) |
| Clinical diagnosis of sepsis prior to onset of NH§§ | | 34 (26.4) |

Data shown in number and percentage (%) unless stated otherwise.

*Gestational age: based on first trimester ultrasound.[25]

†Birthweight: weight measurement considered valid if done in the first 72 hours of life.

‡Small for gestational age: defined as birth weight below the 10th percentile for gestational age and sex, calculated with the INTERGROWTH-21st newborn size application tool (https://intergrowth21.tghn.org).

§Poor start to life: Apgar score<7 at 5 min and/or clinical suspicion of meconium aspiration and/or resuscitation at birth with at least five inflation breaths.

¶Extreme NH categories: (1) two consecutive SBR measurements above the ET threshold of the NICE guideline, (2) SBR levels rising faster than 8.5 µmol/L/hour in combination with one SBR measurement above the ET threshold or with clinical symptoms of acute bilirubin encepalopathy, and (3) a clinical diagnosis of ABE.

**Potential blood group ABO incompatibility: mother–foetus pairs with mother blood group 'O' and neonate blood group 'A' or 'B'. Rhesus factor and Coombs test were not available.

††Diagnosed by G6PD fluorescent test.[22]

‡‡Polycythaemia: two consecutive haematocrit values>70% from a capillary sample or a diagnosis of polycythaemia recorded in the clinical chart.

§§Clinical diagnosis of sepsis: sepsis reported as clinical diagnosis treated at least 5 days by intravenous antibiotics (blood culture confirmations were not available).

ABE, acute bilirubin encepalopathy; ET, exchange transfusion; G6PD, glucose-6-phosphate dehydrogenase; NH, neonatal hyperbilirubinaemia; NICE, National Institute for Health and Clinical Excellence; SBR, serum bilirubin; SMRU, Shoklo Malaria Research Unit.

**Table 2** Neonatal and maternal characteristics of extreme NH survivors present for neurodevelopmental assessment (n=37) and those unavailable or untraceable (n=78)

| Characteristics | Present (n=37) | Untraceable (n=78) | P value |
|---|---|---|---|
| Gender (male) | 20 (54.1) | 47 (60.3) | 0.529 |
| Year of birth (2012–2014) | 24 (64.9) | 41 (52.6) | 0.214 |
| Refugee status | 23 (62.1) | 54 (69.2) | 0.452 |
| First born | 20 (54.1) | 44 (56.4) | 0.812 |
| EGA (weeks+days), median (IQR) | 37+4 (34+5–38+3) | 37+4 (35+1–38+6) | 0.578 |
| EGA<35 weeks | 11 (29.7) | 18 (23.1) | 0.443 |
| Small for gestational age | 5/35 (14.3) | 14/75 (18.7) | 0.571 |
| Normal vaginal delivery | 31 (83.8) | 70 (89.7) | 0.361 |
| Days in SCBU, median (IQR) | 8 (5–16) | 7 (4–13) | 0.201 |
| Clinical diagnosis of sepsis prior to onset of NH | 8 (21.6) | 18 (23.1) | 0.862 |
| Maternal age, median (IQR) | 24 (20–30) | 22 (19–28) | 0.391 |
| Maternal illiteracy | 9/27 (33.3) | 23/57 (40.4) | 0.536 |
| Maternal smoking | 7 (18.9) | 11 (14.1) | 0.525 |

Data shown in number and percentage (%) unless stated otherwise.
EGA, estimated gestational age; NH, neonatal hyperbilirubinaemia; SCBU, special care baby unit.

one hospitalisation per year. The proportion of chronic malnutrition was high (40.5% of children stunted), and half of the participants had teeth damage with black staining or cavities. Abnormal visual examination was reported in 21.6% of children, mostly related to poor acuity. None of the children wore glasses. Overall, children born <35 weeks' EGA were reporting similar, if not slightly better clinical condition than children born term (differences non-significant).

Two survivors of extreme NH (5.4%), both born at term, had an abnormal neurological examination compatible with a severe form of KSD: one with upward and downward gaze paralysis, dystonia and suspected left-sided hearing loss, the other with cerebral palsy. Their general neurodevelopment score was below the first percentile.

The median (IQR) percentile summary score of all GMDS-ER subscales was low (11 (2–42)), with 16 children (43.2%) scoring below the 10th centile; scores were particularly poor in the subscales 'hearing and language', 'performance' and 'practical reasoning' (table 3). Differences observed between gestational age groups did not reach significance level; however, children born <35 weeks' EGA tended to obtain similar or higher scores than those born term. Four survivors of extreme NH (10.8%) had a general percentile score above the 50th centile, the highest score reaching the 82nd centile.

## DISCUSSION

This study reports the immediate outcomes and long-term consequences of extreme NH in a limited-resource setting. Reflecting on the burden estimates for South-East Asia of 251.3 per 10 000 live live births and the worldwide 1309 deaths per 100 000 live births presented in the publications by Slusher[8] and Olusanya[9] both the incidence of

extreme NH and its mortality rate during this 6 year evaluation are reassuringly low. This likely reflects the benefit of onsite SBR measurements and thus an early phototherapy initiation. The case fatality rate among neonates with extreme NH treated with phototherapy was similar to that described in tertiary paediatric referral hospitals in Myanmar.[31]

Clinical symptoms compatible with ABE were present in 28.2% of neonates, a proportion within the range reported in the literature[3 32] but with deleterious consequences: all deaths occurred among neonates with ABE and of the two children with severe neurological impairment at least one was suspected to be ABE sequelae. Routine recording of clinical symptoms using a system such as the Bilirubin-Induced Neurological Dysfunction scales might help capture earlier neurological abnormalities and enhance closer monitoring alongside the use of dynamic treatment thresholds.[21] This, however, could only be of added benefit if timely referral can be organised as well. As a first step, emphasising the need for systematic SBR monitoring, ensuring regular feeds during phototherapy, and verifying that adequate irradiance levels are provided, are achievable recommendations to minimise ABE risk.[18 33]

The one-off neurodevelopmental evaluation results ranged from severely impaired to developmentally normal consistent with current published rates.[7] Test performance was particularly poor in domains related to language and reasoning skills, both plausibly affected by bilirubin neurotoxicity.[6 7 34] Reassuringly, survivors of extreme NH <35 weeks' EGA performed similarly or even better than those born term despite a higher risk of developing ABE.

Whether children with severe KSD retain a normal intelligence is debated.[1 6 35] As an anecdotal case, and

**Table 3** Long-term clinical, visual, neurological and neurodevelopmental outcomes of 37 survivors of extreme NH by gestational age category

| | Gestational age <35 weeks (n=11) | Gestational age ≥35 weeks (n=26) | All (n=37) |
|---|---|---|---|
| **General characteristics** | | | |
| Age at testing (months) (median, IQR) | 62.5 (36.5–93.0) | 43.8 (34–64.5) | 51.0 (35–81) |
| Age <4 years old | 3 (27.3) | 13 (50.0) | 16 (43.2) |
| Gender (male) | 4 (36.4) | 16 (61.5) | 20 (54.1) |
| First born | 8 (72.7) | 12 (46.2) | 20 (54.1) |
| Received exchange transfusion | 0 | 3 (11.5) | 3 (8.1) |
| Refugee status | 7 (63.6) | 16 (61.5) | 23 (62.2) |
| **Clinical outcomes** | | | |
| Being unwell* | 3 (27.3) | 8/25 (32.0) | 11/36 (30.6) |
| Stunting† | 2 (18.2) | 13 (50.0) | 15 (40.5) |
| Teeth abnormality‡ | 5/10 (50.0) | 13/26 (50.0) | 18/36 (50.0) |
| **Visual and neurological outcomes** | | | |
| Visual examination abnormality§ | 1 (9.1) | 7 (26.9) | 8 (21.6) |
| Neurological exam abnormality¶ | 0 | 2 (7.7) | 2 (5.4) |
| **Neurodevelopmental outcomes** | | | |
| Maternal perception of child's development: mother has worries | 3 (27.3) | 3 (11.5) | 6 (16.2) |
| General GMDS percentile score, median (IQR) | 32 (11–42) | 7 (<1–42) | 11 (2–42) |
| Poor development (<10th centile) | 2 (18.2) | 14 (53.8) | 16 (43.2) |
| Delayed development (10–49th centile) | 7 (63.6) | 10 (38.5) | 17 (46.0) |
| Normal development (≥50th centile) | 2 (18.2) | 2 (7.7) | 4 (10.8) |
| Locomotion score, median (IQR) | 34 (19–59) | 12 (3–58) | 19 (5–58) |
| Personal and social score, median (IQR) | 23 (8–94) | 32 (15–64) | 26 (11–64) |
| Hearing and language score, median (IQR) | 12 (6–57) | 11 (<1–33) | 11 (<1–33) |
| Eye and hand coordination score, median (IQR) | 27 (4–46) | 14 (4–48) | 22 (4–46) |
| Performance score, median (IQR) | 26 (6–80) | 3 (<1–19) | 4 (<1–40) |
| Practical reasoning score, median (IQR)** | 18 (2–51) | 11 (1–43) | 14 (2–49) |

Data shown in number and percentage (%).
*Being unwell: more than one hospitalisation per year or presence of a chronic disease as reported in the child's medical booklet.
†Stunting: height-for-age z-scores below −2 as calculated with WHO Child Growth Standards (http://www.who.int/childgrowth/en/).
‡Teeth abnormality: judgement of medical staff looking for the presence of black staining or cavities, confirmed by first author through photographic records.
§Visual examination abnormality: unconjugated eye movement and Cardiff's binocular visual contrast and visual acuity lower than the normal values by age categories (instructions manual for the Cardiff Acuity Test).
¶Neurological examination abnormality: judgement of the medical staff after general observation while moving, and evaluating tone, strength and sensibility, coordination, gait and posture, and reflexes.
**One child had an adjusted age of <2 years old when tested; therefore, practical reasoning was available for only 10 preterm children.
EGA, estimated gestational age; GMDS, Griffiths Mental Development Scale; NH, neonatal hyperbilirubinaemia; SCBU, special care baby unit.

even though it was not possible to measure intelligence, the single child with features compatible with classic kernicterus appeared frustrated with not being able to complete the proposed tasks.

The clinical findings of this study confirm the harsh environment in which the children try to thrive: a high burden of chronic malnutrition, infections, and poor visual and dental care.[36–40] Moderate to severe delay in development after surviving extreme NH will add to an already heavy burden and to the worries of families. In

such a precarious socioeconomic environment, caregivers of neonates with extreme NH should be encouraged to attend a baby clinic to actively identify and treat nutritional and clinical problems influencing growth and development, if this is available.

This study's results might not be extrapolated elsewhere due to its limitations. ABE symptoms thought to represent abnormal neurological conditions were extracted from the daily medical notes and the lack of a systematic checklist means that the proportion of ABE might have

been underestimated. The absence of regular follow-up after discharge from the SCBU limited the search for survivors 2–8 years later to those families who remained in the area and possibly underestimated the mortality rate while overestimating the proportion of children with severe delayed development if one assumes the family would remain closest to what little healthcare services are available. Neurodevelopmental tests were performed by locally trained staff fluent in local languages rather than by specialists, and cultural and environmental factors could have influenced the overall poor test outcomes. Nevertheless, all children came from the same environment; school exposure was generally limited; and the health staff was routinely quality controlled to report the pass/fail items as per test guideline and was unaware of the final scoring.

## CONCLUSION

Extreme NH is a serious neonatal health issue in this limited-resource setting and possibly contributes to delayed development of children already vulnerable to a heavy burden of chronic malnutrition and infections. Early identification and adequate treatment of NH where ET is not readily available are key to minimising the risk of extreme SBR values or neurological symptoms; additional efforts are necessary to encourage parents to attend the baby clinic to identify and treat early concomitant health and nutritional problems.

**Author affiliations**
[1]Shoklo Malaria Research Unit, Mahidol-Oxford Tropical Medicine Research Unit, Faculty of Tropical Medicine, Mahidol University, Mae Sot, Thailand
[2]University Medical Center, University of Groningen, Groningen, Netherlands
[3]Neonatology-Pediatrics Department, Hôpital Erasme, Université Libre de Bruxelles, Bruxelles, Belgium
[4]Faculty of Tropical Medicine, Mahidol-Oxford Tropical Medicine Research Unit, Mahidol University, Bangkok, Thailand
[5]IRD-INSERM-SESSTIM, Aix-Marseille Université, Marseille, France
[6]Paediatric Gastroenterology, University Medical Center Groningen, Groningen, Netherlands
[7]Cambodia-Oxford Medical Research Unit, Angkor Hospital for Children, Siem Reap, Cambodia
[8]Centre for Tropical Medicine and Global Health, Nuffield Department of Medicine, University of Oxford, Oxford, UK
[9]Department of Medicine, Swiss Tropical and Public Health Institute, Basel, Switzerland

**Acknowledgements** We thank all the SMRU staff and the parents and children who were willing to join in the study. Thanks to Stepping Stones physiotherapy for caring for the disabled children.

**Contributors** EMNW, LT and VIC conceived and designed the study, coordinated the study and supervised the research teams; PFvR, CT, FN and RMcG contributed to the study design. MCD, AAN, WS and SK performed all the neurodevelopment tests and clinical examinations. TN developed the data collection and provided the quality control. EMNW, LT and VIC led the analysis and interpretation of the data with JL leading the statistical analysis. All authors supported the preparation and later critically reviewed the manuscript, including in its final approval, and had full access to all of the data (including statistical reports and tables) in the study and can take responsibility for the integrity of the data and the accuracy of the data analysis.

**Funding** Shoklo Malaria Research Unit is part of the Mahidol-Oxford University Tropical Medicine Research Program at the Faculty of Tropical Medicine, Mahidol University, Bangkok, Thailand, and supported by the Wellcome Trust of Great Britain. The funders had no role in study design, data collection and analysis, decision to publish or preparation of the manuscript. LT was supported by a PhD grant from 'The Belgian Kids' Fund for Pediatric Research.

**Map disclaimer** The depiction of boundaries on the map(s) in this article does not imply the expression of any opinion whatsoever on the part of BMJ (or any member of its group) concerning the legal status of any country, territory, jurisdiction or area or of its authorities. The map(s) are provided without any warranty of any kind, either express or implied.

**Competing interests** None declared.

**Patient consent for publication** Not required.

**Ethics approval** The study was approved by the Oxford Tropical Research Ethics Committee (OXTREC 5113–16) and the ethics committee of the Faculty of Tropical Medicine of the Mahidol University (TMEC 16–071). The Tak Province Border Community Ethics Advisory Board representing the local community reviewed and accepted the study protocol (TCAB-01/REV/2016).

**Provenance and peer review** Not commissioned; externally peer reviewed.

**Data availability statement** Datasets used for this analysis are available from the corresponding author upon reasonable request.

**ORCID iDs**
Eva Maria Nadine Wouda http://orcid.org/0000-0002-7192-3422
Verena Ilona Carrara http://orcid.org/0000-0002-2758-0872

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
