## [Reviewer comments · BMJ Paediatrics Open]

ARTICLE DETAILS

TITLE (PROVISIONAL)	EXTREME NEONATAL HYPERBILIRUBINEMIA IN REFUGEE AND MIGRANT POPULATIONS: RETROSPECTIVE COHORT
AUTHORS	Wouda, Eva; Thielemans, Laurence; Darakamon, Mue; Nge, Aye; Say, Wah; Khing, Sanda; Hanboonkunupakarn, Borimas; Ngerseng, Thatsanun; Landier, Jordi; van Rheenen, Patrick; Turner, Claudia; Nosten, Francois; McGready, Rose; Carrara, Verena

VERSION 1 – REVIEW

REVIEWER	Reviewer name: Tina Slusher Institution and Country: University of Minnesota and Hennepin Healthcare. USA Competing interests: None
REVIEW RETURNED	29-Jan-2020

GENERAL COMMENTS	This is an extremely important article adding to the much needed information re severe neonatal jaundice/hyperbilirubinemia and is well written. You appropriately point out likely reasons your results were better than expected and your conclusion is "spot on". However, there are a few additions/ revisions that would improve this paper. The authors need to explain why they chose the Griffiths Mental Development score to the audience i.e. why they believe it is valid in their population (low-resource with potential for kernicterus/kernicterus spectrum disorder) and how well it measures actual IQ. Many authors including well known experts in the fields (Shapiro and Volpe) believe many if not most of these children have normal to high IQ and are "trapped in a body that does not work" and certainly in this reviewer's personal experience that has often been the case. Would the Griffiths Mental Development score demonstrate this if this were the case in your children at follow-up? I would also suggest using the term Kernicterus Spectrum Disorder as described by Le Pichon et al as this more accurately describes the chronic sequel of acute bilirubin encephalopathy as it is indeed a spectrum.
--

REVIEWER	Reviewer name: Gaston Arnolda Institution and Country: Macquarie University, Australia Competing interests: None
REVIEW RETURNED	10-Feb-2020

GENERAL COMMENTS	Background The manuscript addresses aspects of an important ongoing problem in neonatal care in low resource settings: the prevalence and consequences of extreme neonatal hyperbilirubinaemia ('ENH'). The data reflects experience on the Thai-Myanmar border, in a clinical setting where exchange transfer is not routinely available; a setting from which data are rarely available. I thank the authors for their efforts in seeking to bring their experiences to a broader audience.
--

The paper appears to seek to achieve two distinct objectives: 1) to estimate the incidence of ENH in a cohort of livebirths and describe the characteristics of infants with ENH; and 2) to compare Griffith scores for a sub-sample of the cases with ENH with a selection of controls, as an indication of the consequences of ENH.

Overall summary of reviewer's opinion

It appears likely that Objective 1 can be achieved through clarification of, or provision of additional information in, the manuscript. Given the setting, and the rarity of data from it, I believe that provision of this data is sufficient to justify publication, though I leave it to the editors to determine the relevance of this type of information to their readership.

On the information available in the manuscript, it is unclear to me whether Objective 2 can be achieved, because the cases were matched for year of birth but not for gestational age. There is an important difference in the proportion of cases and controls born <35 weeks estimated gestation (30% vs 10%, respectively), and it is therefore unclear how much of the difference in average Griffith scores are attributable to prematurity rather than ENH. One possible way forward is to separately (or only) report data for cases aged 35 weeks and above, and their age-matched controls; I appreciate that this further reduces an already small sample size, but believe it would give the reader greater confidence in the findings.

Alternatively, one could just focus on describing the Griffith scores for the cases.

My comments below focus on bigger picture issues only; if the editors invite the authors to re-submit, I would be happy to review the revision in additional detail.

Specific issues relating to Objective 1

Below are a few key issues that appeared to me to require clarification:

1. Inborn vs Outborn: The data in Table 1 appears to conflate data for inborn and outborn infants (e.g., the 129 ENH cases includes 4 cases born in a tertiary hospital and 15 born at home). For the sake of consistency and accurate calculation of the incidence of ENH, I would suggest restricting the study to infants born at an SMRU clinic. This would of course mean reduction of numbers with ENH (and likely reduction of cases and matched controls). It is of course possible that this modification would distort the situation if, for example, the tertiary births were antenatal referrals from SMRU and/or if SMRU had reduced the number of home births over time, by actively promoting facility birth; if so, this needs to be explained and justified so that it can be understood.

2. Site-specific data: The data are sourced from three sites, one contributing data from 2009-2014 with two more added from 2011-2014. For the most part the data are aggregated and reported as if for a single site. It would be useful to have additional information by site, to better understand the context. For example, Figure 2 could be extensively modified to provide the data separately by site. Such information may help to explain why the number of ENH cases remains roughly the same for each year over the 2009-2014 period except for a low number in 2011; Table 1), despite the fact that two new clinics were added (e.g., did these new sites merely take births from the older site?). If births did increase markedly, we need to know why the rate of ENH appears to have reduced (e.g., improved monitoring and treatment, better quality phototherapy etc). [As an aside, restricting data to SMRU births would permit a tracking of incidence of ENH over time, to allow assessment of time-trend]

3. Quality of phototherapy available:

	As the authors note in the discussion, the quality of phototherapy treatment is crucial, especially in the absence of exchange transfusion. It would be useful to know about the quality of phototherapy available, how it was assured (e.g., long-lived LED, frequency of measurement of irradiance if using fluorescent), and whether it changed at any of the sites during the study period. 4. Definition of ENH: The definition of ENH appears one of three criteria: i) ET threshold in NICE; ii) SBR rising above a specified rate + other criteria; iii) a clinical diagnosis of ABE. It would be useful to know, in Table 1, the number of infants qualifying by each of these criteria. As background, it would also be useful to know the way each of the SMRU clinics measured SBR, and how (and how often) this was quality assured. Specific issues relating to Objective 2 Below are a few key issues that appeared to me to require clarification or modification:  1. Controls: As I read it, the controls were selected only as being born in the same year; if there was additional matching (e.g., by SMRU site), this needs to be clearly specified. 2. Respondent vs non-respondent cases: Given the low rate of case follow-up assessment (37/113 not known to have died), I believe detailed reporting comparing respondents and non-respondents is essential, focussing on the potential clinical implications of differences between the groups, rather than statistical significance. If the information cannot fit within the word limits, I recommend that it be included in an Appendix. 3. Effect of gestation on Griffiths scores: As indicated earlier, the excess of infants born <35 weeks gestation among cases makes it difficult to assess the contribution of ENH on Griffiths scores. One option is to drop all cases born <35 weeks (and their associated controls); this implies that if a mildly preterm or term case only has controls born <35 weeks, the case would also need to be dropped. In my view the loss of statistical power is more than compensated by the reduction in potential for confounding. Another alternative is to drop the control series altogether, for the purposes of this manuscript, and to restrict your discussion to a description of the cases, perhaps with some reformatting of tables to report Griffiths scores by gestational sub-group. Final comments Having previously worked with clinicians in low-resource settings, I appreciate how much hard work has gone into collecting, cleaning and analysing the data. I hope my suggestions will help to improve the value of your work to others interested in care of neonates in these challenging settings. Thank you for the opportunity to comment on your work.
--	---

VERSION 1 – AUTHOR RESPONSE

Reviewer: 1

This is an extremely important article adding to the much needed information re severe neonatal jaundice/hyperbilirubinemia and is well written. You appropriately point out likely reasons your results were better than expected and your conclusion is "spot on". However, there are a few additions/ revisions that would improve this paper.

The authors need to explain why they chose the Griffiths Mental Development score to the audience i.e. why they believe it is valid in their population (low-resource with potential for kernicterus/kernicterus spectrum disorder) and how well it measures actual IQ.

Many authors including well known experts in the fields (Shapiro and Volpe) believe many if not most of these children have normal to high IQ and are "trapped in a body that does not work" and certainly in this reviewer's personal experience that has often been the case. Would the Griffiths Mental Development score demonstrate this if this were the case in your children at follow-up?

Answer:

We appreciate this comment and thank you for the advice. The Griffiths Mental Development Scales was primarily selected because it spreads across the age spectrum we were interested in and because the test procedure and items were already known by the locally trained neurodevelopment testers; in addition, a selection of the test items are included in validated tests that are used daily in this setting (Haataja L: *Annals of tropical paediatrics*. 2002;22(4):355-6 and Abubakar A: *Annals of tropical paediatrics*. 2008;28(3):217-26.).

The GMDS is not indeed a test that can evaluate Intellectual Quotient, but it nevertheless allows the evaluation of a wide range of delays; ie. motor, verbal, or associated to executive function all potentially associated with neurological sequelae of bilirubin brain toxicity. As for the two children with severe motor and verbal impairment they both scored extremely poorly on the test although the personal feeling of the tester was that the children were willing to accomplish the task, frustrated even as their disabilities were "locking them in".

We have added a paragraph in the methods to explain our choice of test (lines 130-133) and completed the discussion (lines 225-228).

I would also suggest using the term Kernicterus Spectrum Disorder as described by Le Pichon et al as this more accurately describes the chronic sequel of acute bilirubin encephalopathy as it is indeed a spectrum.

Answer:

We have amended the abstract and text as suggested and added the reference.

Reviewer: 2

Background

The manuscript addresses aspects of an important ongoing problem in neonatal care in low resource settings: the prevalence and consequences of extreme neonatal hyperbilirubinaemia ('ENH'). The data reflects experience on the Thai-Myanmar border, in a clinical setting where exchange transfer is not routinely available; a setting from which data are rarely available. I thank the authors for their efforts in seeking to bring their experiences to a broader audience.

The paper appears to seek to achieve two distinct objectives: 1) to estimate the incidence of ENH in a cohort of livebirths and describe the characteristics of infants with ENH; and 2) to compare Griffith scores for a sub-sample of the cases with ENH with a selection of controls, as an indication of the consequences of ENH.

Overall summary of reviewer's opinion

It appears likely that Objective 1 can be achieved through clarification of, or provision of additional information in, the manuscript. Given the setting, and the rarity of data from it, I believe that provision of this data is sufficient to justify publication, though I leave it to the editors to determine the relevance of this type of information to their readership.

On the information available in the manuscript, it is unclear to me whether Objective 2 can be achieved, because the cases were matched for year of birth but not for gestational age.

There is an important difference in the proportion of cases and controls born <35 weeks estimated gestation (30% vs 10%, respectively), and it is therefore unclear how much of the difference in average Griffith scores are attributable to prematurity rather than ENH. One possible way forward is to separately (or only) report data for cases aged 35 weeks and above, and their age-matched controls; I appreciate that this further reduces an already small sample size, but believe it would give the reader greater confidence in the findings. Alternatively, one could just focus on describing the Griffith scores for the cases.

My comments below focus on bigger picture issues only; if the editors invite the authors to re-submit, I would be happy to review the revision in additional detail.

Specific issues relating to Objective 1

Below are a few key issues that appeared to me to require clarification:

1. Inborn vs Outborn: The data in Table 1 appears to conflate data for inborn and outborn infants (e.g., the 129 ENH cases includes 4 cases born in a tertiary hospital and 15 born at home). For the sake of consistency and accurate calculation of the incidence of ENH, I would suggest restricting the study to infants born at an SMRU clinic. This would of course mean reduction of numbers with ENH (and likely reduction of cases and matched controls). It is of course possible that this modification would distort the situation if, for example, the tertiary births were antenatal referrals from SMRU and/or if SMRU had reduced the number of home births over time, by actively promoting facility birth; if so, this needs to be explained and justified so that it can be understood.

Answer:

Thank you to have pointed out this inaccuracy; all children born in a tertiary hospital are indeed referrals from SMRU a number which has remained constant through the years (7.7%) and we have reduced home birth over time as you pointed out (from 18% to 9%). The incidence has been recalculated without neonates born outside SMRU facilities (lines 156-158); however we decided to keep the total number of 129 neonates diagnosed with extreme NH at SMRU for the rest of the analysis; only a minority of neonates born at home are never seen again at SMRU (<2%) and all the antenatal referrals to tertiary hospital are sent back to SMRU clinics at discharge and all the mothers have attended antenatal care at SMRU; furthermore the description of the cohort itself is independent of the incidence data.

2. Site-specific data: The data are sourced from three sites, one contributing data from 2009-2014 with two more added from 2011-2014. For the most part the data are aggregated and reported as if for a single site. It would be useful to have additional information by site, to better understand the context. For example, Figure 2 could be extensively modified to provide the data separately by site. Such information may help to explain why the number of ENH cases remains roughly the same for each year over the 2009-2014 period except for a low number in 2011; Table 1), despite the fact that two new clinics were added (e.g., did these new sites merely take births from the older site?). If births did increase markedly, we need to know why the rate of ENH appears to have reduced (e.g., improved monitoring and treatment, better quality phototherapy etc). [As an aside, restricting data to SMRU births would permit a tracking of incidence of ENH over time, to allow assessment of time-trend]

Answer:

The relatively stable numbers of extreme NH cases over the years despite an increase number of settings is indeed due to an improved monitoring, earlier care and adequate phototherapy irradiance which happened as the new clinics were opening; the role of improved monitoring and care and its effect on incidence over time has already been published by Thielemans et al. (BMC Pediatrics, 2018;

18:190) and therefore only minimal information has been added to this paper focusing on describing the clinical and developmental consequences of extreme NH. We have added some clarification in the introduction (lines 81-84).

3. Quality of phototherapy available: As the authors note in the discussion, the quality of phototherapy treatment is crucial, especially in the absence of exchange transfusion. It would be useful to know about the quality of phototherapy available, how it was assured (e.g., long-lived LED, frequency of measurement of irradiance if using fluorescent), and whether it changed at any of the sites during the study period.

Answer:

Effectively the quality of the phototherapy improved over time with the purchase of several additional LED units; this has likely contributed to the overall number of extreme NH cases; this is part of the publication mentioned previously (Thielemans et al 2018); we have added some clarification in the text within the limits of the words permitted (lines 81-84).

4. Definition of ENH: The definition of ENH appears one of three criteria: i) ET threshold in NICE; ii) SBR rising above a specified rate + other criteria; iii) a clinical diagnosis of ABE. It would be useful to know, in Table 1, the number of infants qualifying by each of these criteria. As background, it would also be useful to know the way each of the SMRU clinics measured SBR, and how (and how often) this was quality assured.

Answer:

The number of neonates by extreme NH criteria of selection has been added to Table 1. The method of SBR measurements has been added to the text (lines 81-84). All laboratory tests were performed by trained and quality controlled laboratory technicians (lines 121-122); equipment maintenance and calibration was performed routinely as per laboratory SOPs.

Specific issues relating to Objective 2

Below are a few key issues that appeared to me to require clarification or modification:

1. Controls: As I read it, the controls were selected only as being born in the same year; if there was additional matching (e.g., by SMRU site), this needs to be clearly specified.

Answer:

We indeed attempted to have a comparative group of children not only born the same year but also with a similar distribution in gender and site (refugee camp versus migrant communities); as pointed out in later comments, obtaining a comparative distribution of gestational age was indeed near impossible as neonates born less than 35 weeks of gestational age nearly all developed jaundice severe enough to be excluded from the comparative group.

However, as suggested by you and by the editor-in-chief, we have removed the comparison with those children.

2. Respondent vs non-respondent cases: Given the low rate of case follow-up assessment (37/113 not known to have died), I believe detailed reporting comparing respondents and non-respondents is essential, focussing on the potential clinical implications of differences between the groups, rather than statistical significance. If the information cannot fit within the word limits, I recommend that it be included in an Appendix.

Answer:

The general neonatal and familial characteristics of both respondent and non-respondents children are now presented in table 2.

3. Effect of gestation on Griffiths scores: As indicated earlier, the excess of infants born <35 weeks gestation among cases makes it difficult to assess the contribution of ENH on Griffiths scores. One option is to drop all cases born <35 weeks (and their associated controls); this implies that if a mildly preterm or term case only has controls born <35 weeks, the case would also need to be dropped. In my view the loss of statistical power is more than compensated by the reduction in potential for confounding. Another alternative is to drop the control series altogether, for the purposes of this manuscript, and to restrict your discussion to a description of the cases, perhaps with some reformatting of tables to report Griffiths scores by gestational sub-group.

Answer:

Gestational age was accounted for while calculating the scores of children born prematurely as per manual guidelines (lines 134-136).

The table 3 now presents the Griffiths scores of the 37 survivors by gestational age category as suggested. The results (lines 175-198) and discussion (lines 219-228) have been adapted accordingly.

Final comments

Having previously worked with clinicians in low-resource settings, I appreciate how much hard work has gone into collecting, cleaning and analysing the data. I hope my suggestions will help to improve the value of your work to others interested in care of neonates in these challenging settings. Thank you for the opportunity to comment on your work.